# Deep Energy-Based Modeling of Discrete-Time Physics

**Takashi Matsubara**
Osaka University
Osaka, Japan 560–8531
matsubara@sys.es.osaka-u.ac.jp

**Ai Ishikawa**
Kobe University
Kobe, Japan 657–8501
a-ishikawa@stu.kobe-u.ac.jp

**Takaharu Yaguchi**
Kobe University
Kobe, Japan 657–8501
yaguchi@pearl.kobe-u.ac.jp

## Abstract

Physical phenomena in the real world are often described by energy-based modeling theories, such as Hamiltonian mechanics or the Landau theory, which yield various physical laws. Recent developments in neural networks have enabled the mimicking of the energy conservation law by learning the underlying continuous-time differential equations. However, this may not be possible in discrete time, which is often the case in practical learning and computation. Moreover, other physical laws have been overlooked in the previous neural network models. In this study, we propose a deep energy-based physical model that admits a specific differential geometric structure. From this structure, the conservation or dissipation law of energy and the mass conservation law follow naturally. To ensure the energetic behavior in discrete time, we also propose an automatic discrete differentiation algorithm that enables neural networks to employ the discrete gradient method.

## 1 Introduction

Deep neural networks have achieved significant results for a variety of real-world tasks such as image processing [23, 46], natural language processing [13], and game playing [40]. Their successes depend on hard-coded prior knowledge, such as translation invariance in image recognition [28] and the manifold hypothesis in data modeling [36]. The prior knowledge guarantees a desirable property of the learned function. The Hamiltonian neural network (HNN) [19] implements the Hamiltonian structure on a neural network and thereby produces the energy conservation law in physics. After its great success, neural networks specifically designed for physical phenomena have received much attention. They have been intensively extended to various forms, such as the Hamiltonian systems with additional dissipative terms [44].

Meanwhile, most previous studies aimed to model continuous-time differential equations and employed numerical integrators (typically, an explicit Runge–Kutta method) to integrate the neural network models for learning and computing the dynamics [7, 8, 19, 45]. Surprisingly, our numerical experiments reveal that a higher-order numerical integrator with adaptive time-stepping is quite often inferior in performance as compared to a quantitatively lower order but qualitatively superior numerical integrator. This is because higher-order integrators aim to reproduce continuous-time dynamics while practical learning and computation are in discrete time. In this case, the qualitative features that the integrators equipped with could be actually essential.

From this point of view, this study proposes a *deep energy-based discrete-time physical model*, which combines neural networks and discrete-time energy-based modeling. The key ingredient is the structure-preserving integrators, in particular, the discrete gradient method along with the newly-developed automatic discrete differentiation. In addition, our framework unifies and also extends the aforementioned previous studies. The main contributions include:

**Applicable to general energy-based physical models.** Our framework is applicable to general physical phenomena modeled by the energy-based theory, such as Hamiltonian mechanics, the Landau theory, and the phase field modeling. Our target class includes a Hamiltonian system composed of position and momentum (a so-called natural system, such as a mass-spring system), a natural system with friction, a physical system derived from free-energy minimization (e.g., phase transitions), and a Hamiltonian partial differential equation (PDE) (e.g., the Korteweg–de Vries (KdV) equation and the Maxwell equation). All equations can be written as a geometric equation. Most studies have focused on one of the first two systems [19, 44, 45] under special conditions [8, 38, 41], or they are too general to model the conservation and dissipation laws [7, 34]. The details of the proposed framework along with the target class of the equations and the geometric aspects are described in Section 3.1.

**Equipping with the laws of physics in discrete time.** Previous models interpolate the discrete-time data using numerical integrators for learning and computing [8, 19, 38, 41, 44, 45]. The discretization may destroy the geometrical structure from which the laws of physics follow (see the lower part of Fig. 1). Conversely, our approach, in principle, learns a discrete-time model from the discrete-time data without the time-consuming interpolation and discretization error (see

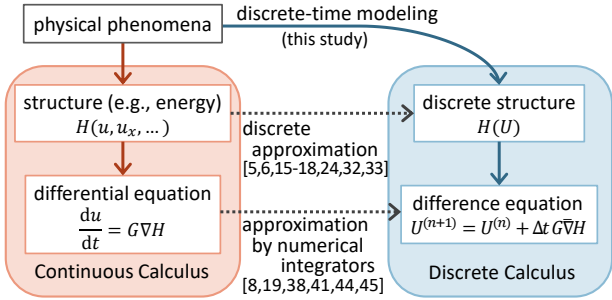

Figure 1: Modeling based on energy-based theories.

the upper part). Using the *discrete gradient*, our approach admits the important laws of physics, particularly the energy conservation or dissipation law and the mass conservation law in discrete time. We demonstrate this property theoretically in Section 3.2 and experimentally in Section 4.

**Easy-to-use.** Our approach is based on the discrete gradient method [15, 18, 33]. Most discrete gradients require the explicit form of the function (see the middle part of Fig. 1); hence, they are unavailable for neural networks (see Appendix A for reference). We propose an automatic discrete differentiation algorithm, which automatically obtains the discrete gradient of the neural networks composed of linear and nonlinear operations. The proposed algorithm can be implemented in a similar way to the current automatic differentiation algorithm [20]; we provide it as a PyTorch library [31][1]. We introduce the detailed algorithm in Section 3.3.

## 2 Related Work

**Neural Networks for Differential Equations.** Since the 1990s, many studies have attempted to approximate ordinary differential equations (ODEs) and PDEs by applying neural networks [1, 9, 27, 34, 35, 37]. Recent advances in the automatic differentiation algorithm [20] have enabled us to build more complicated neural network architectures. Neural ODE (NODE) [7] has re-established neural networks for modeling ODEs. NODE treats the output of a time-dependent neural network as the time derivative of the input; thereby, defining an ODE in a general way. Moreover, NODE employs numerical integrators to train and integrate the neural network model. Several studies attempted to model a PDE system using regularization terms to mimic the conservation laws [34, 42]. They were insufficient to ensure the conservation laws in physical systems.

The HNN approximates an energy function $H$ from the data using a neural network, and thereby, builds a Hamiltonian system [19]. The time-derivative of the states $(\vec{q}, \vec{p})$ is given using the gradient $\nabla H$ of the energy $H$, which is called the Hamiltonian, specifically, $\mathrm{d}\vec{q}/\mathrm{d}t = \nabla_{\vec{p}}H$ and $\mathrm{d}\vec{p}/\mathrm{d}t =$

Table 1: Comparison with Other Studies

|  | HNN [19] | SymODEN [45] | Dissipative [44] | SRNN/VIN [8, 38, 41] | DGNet (this paper) |
|---|---|---|---|---|---|
| Hamiltonian system | yes | yes | yes | yes | yes |
| Dissipative ODE |  |  | yes |  | yes |
| Hamiltonian PDE |  |  |  |  | yes |
| Dissipative PDE |  |  |  |  | yes |
| Learning from finite difference |  | approx.* | approx.* | approx.* | yes |
| Strict conservation law in discrete-time |  |  |  | approx.** | yes |
| Strict dissipation law in discrete-time |  |  |  |  | yes |

* Interpolating by numerical integrators. ** Conserving only the "shadow" Hamiltonian.

$-\nabla_{\vec{q}} H$, where $\vec{q}$ and $\vec{p}$ denote the position and momentum, respectively. Following the HNN, the symplectic ODE-Net integrates an HNN-like model using a Runge–Kutta method; thus, enabling learning from the discrete-time data [45]. The dissipative SymODEN generalized it to a model with friction and input [44]. We summarized the previous studies in Table 1.

**Structure-Preserving Numerical Methods.** Most differential equations that arise as models for physical phenomena admit some laws of physics, e.g., the energy and other conservation laws of the Hamilton equation and the mass conservation law and energy dissipation properties of the equations for phase-transition phenomena. Numerical integrators that reproduce those properties are called structure-preserving integrators or geometric integrators [22].

The aforementioned studies mainly employed classical Runge–Kutta methods for numerical integration, which in general destroy these properties [22]. Several recent studies have employed symplectic integrators, which conserve an approximated energy called a "shadow" Hamiltonian in discrete time [8, 38, 41]. These studies considered only the systems of which the Hamiltonian $H$ is separable, i.e., expressible as the sum of the potential and kinetic energies. This is quite restrictive; in fact, most of the important Hamiltonian PDEs (e.g., the shallow water equations and the nonlinear Schrödinger equation) are not in this class. Moreover, structure-preserving integrators for dissipative systems have never been employed. This is because these integrators are often based on the discrete gradient method; however, no efficient discrete gradient has been available for neural networks.

Several studies have focused on Lagrangian mechanics [10, 38]. Lagrangian mechanics can be expressed using the time derivative of the position, while the Hamiltonian mechanics requires conjugate momentum. The main drawback is that it is not obviously extendable to general dissipative systems. We consider it out of scope of this study, but the proposed method is extendable to it [43].

## 3 Methods

### 3.1 General Form of Energy-Based Dynamical Systems

We focus on the following formulation of the models by the energy-based theories, which expresses a wide variety of physical systems described by ODEs and discretized PDEs [15, 32]. The system has a state $\vec{u} \in \mathbb{R}^N$ and an energy function $H : \mathbb{R}^N \to \mathbb{R}$. The time evolution is expressed as

$$\frac{\mathrm{d}\vec{u}}{\mathrm{d}t} = G(\vec{u})\nabla H(\vec{u}), \tag{1}$$

where $G \in \mathbb{R}^N \times \mathbb{R}^N$ is a matrix, which can be state-dependent, and $\nabla H(\vec{u})$ is the gradient of the system energy $H$ with respect to the state $\vec{u}$. Systems of this form arise as differential geometric equations on Riemannian or symplectic manifolds. See Appendix B for reference. The system $(H, G, \vec{u})$ has the following laws of physics.

**Theorem 1.** *The system has the energy dissipation law if $G \leq O$ and the energy conservation law if $G$ is skew-symmetric.*

See Appendix C for the proofs of the theorems for this study. Note that $G \leq O$ denotes that the matrix $G$ is negative semi-definite, with which $\vec{x}^\top G \vec{x} \leq 0$ for any vector $\vec{x}$. A matrix $G$ is skew-symmetric if $G^\top = -G$, and then $\vec{x}^\top G \vec{x} = 0$ for any vector $\vec{x}$.

**Theorem 2.** *The system has the mass conservation law in the sense that* $\mathrm{d}(\sum_k u_k)/\mathrm{dt} = 0$ *if the vector* $\vec{1} = (1, 1, \ldots, 1)$ *is in the left kernel of* $G$ *(i.e.,* $\vec{1}G = \vec{0}$*).*

Thus, we can design the neural network models with the above laws of physics by defining the models $(H, G, \vec{u})$, where $G$ satisfies the required conditions for the laws of physics shown in the above theorems and $H$ is designed by a neural network.

**Remark 1.** *The models* $(H, G, \vec{u})$ *with* $H$ *represented by neural networks widely extend the scope of the previous studies. In particular, the discretized-in-space PDEs (e.g., the KdV equation [26] and the Cahn–Hilliard equation [3]) have not been treated like this before. This is a significant contribution in this study.*

A natural system is a Hamiltonian system associated to a Hamiltonian function $H$ that is the sum of the potential and kinetic energies. This is expressed as the system $(H, G = S, \vec{u})$ for the matrix

$$S = \begin{pmatrix} O & I_n \\ -I_n & O \end{pmatrix}, \tag{2}$$

where $2n = N$ and $I_n$ denotes an $n$-dimensional identity matrix. The first $n$ elements of the state $\vec{u}$ denote the position $\vec{q}$ and the remaining denotes the momentum $\vec{p}$. The matrix $S$ is skew-symmetric, and the system $(H, G = S, \vec{u})$ conserves the system energy $H$. A pendulum, a mass-spring system, and N-body problems are expressible by this form. Besides, the system $(H, G = S - R, \vec{u})$ expresses a natural system with friction when $S$ is the one shown above and

$$R = \mathrm{diag}(0 \ldots 0 \; g_1 \ldots g_n), \tag{3}$$

where $g_k \geq 0$ is a friction term that dampens the momentum $p_k$; thus, dissipating the system energy $H$ because $(S - R) \leq O$. Most previous studies focused on these two types of systems [8, 19, 38, 44, 45].

From a geometric point of view, the matrix $G$ in the above form means that the systems are defined on cotangent bundles, while the following approach is formulated on general symplectic or Riemannian manifolds, enabling our method to handle the various PDE systems [22]. In fact, the formulation $(H, G, \vec{u})$ can express the discretized PDE systems. For example, PDEs under the periodic boundary condition can be discretized by using the central difference operators, of which the matrix representations are as follows.

$$D = \frac{1}{2\Delta x} \begin{pmatrix} 0 & 1 & & & -1 \\ -1 & 0 & 1 & & \\ & & \ddots & & \\ & & -1 & 0 & 1 \\ 1 & & & -1 & 0 \end{pmatrix}, \quad D_2 = \frac{1}{(\Delta x)^2} \begin{pmatrix} -2 & 1 & & & 1 \\ 1 & -2 & 1 & & \\ & & \ddots & & \\ & & 1 & -2 & 1 \\ 1 & & & 1 & -2 \end{pmatrix}, \tag{4}$$

where $\Delta x$ is the space mesh size. The matrices $D$ and $D_2$ represent first–order and second–order central differences, respectively. The $k$-th element $u_k$ of the state $\vec{u}$ corresponds to the mass at the position $x = k\Delta x$, and the systems $(H, G = D, \vec{u})$ and $(H, G = D_2, \vec{u})$ admit the mass conservation law. For suitable discretization of general differential operators, see Appendix D. The system $(H, G = D, \vec{u})$ is a Hamiltonian PDE, which includes the shallow water equations such as the KdV equation, the advection equation, and the Burgers equation [2]. The matrix $D$ is skew-symmetric; hence, the system $(H, G = D, \vec{u})$ conserves the energy $H$. The system $(H, G = D_2, \vec{u})$ expresses a physical system derived from the Landau free-energy minimization including the Cahn–Hilliard equation and the phase–field model for the phase transitions and the pattern formulations. The energy $H$ dissipates because $D_2 \leq O$. Other target equations include the equations with complex state variables, such as the Schrödinger equation and the Ginzburg–Landau equation. See [15] for details.

## 3.2 Discrete Gradient for Energetic-Property-Preserving Integration

The discrete gradient is defined as the following vector-valued function [15, 18, 24, 33].

**Definition 1.** *For* $H : \mathbb{R}^N \to \mathbb{R}$, $\overline{\nabla}H : \mathbb{R}^N \times \mathbb{R}^N \to \mathbb{R}^N$ *that satisfies the following conditions is called a discrete gradient of* $H$:

$$H(\vec{u}) - H(\vec{v}) = \overline{\nabla}H(\vec{u}, \vec{v}) \cdot (\vec{u} - \vec{v}), \quad \overline{\nabla}H(\vec{u}, \vec{u}) = \nabla H(\vec{u}), \tag{5}$$

*where* $\cdot$ *denotes an inner product.*

The first condition corresponds to the chain-rule $\mathrm{d}H(\Delta\vec{u}; \vec{u}) = \nabla H(\vec{u}) \cdot \Delta\vec{u}$ for the Fréchet derivative $\mathrm{d}H(\cdot; \vec{u})$ of $H$ at $\vec{u}$ and an infinitesimal change $\Delta\vec{u}$ of $\vec{u}$. The second condition verifies that the discrete

gradient $\overline{\nabla} H$ is certainly an approximation of the gradient $\nabla H$. The inner product is typically the standard Hermitian inner product for ODEs and the discrete $L^2$ inner product $\langle \vec{u}, \vec{v} \rangle_{L_{\mathrm{d}}^2} := \sum u_k v_k \Delta x$ for discretized PDEs.

With the discrete gradient $\overline{\nabla} H$, a discrete analogue of the system in Eq. (1) is expressed as follows.

$$\frac{\vec{u}^{(n+1)} - \vec{u}^{(n)}}{t^{(n+1)} - t^{(n)}} = \overline{G}(\vec{u}^{(n+1)}, \vec{u}^{(n)}) \overline{\nabla} H(\vec{u}^{(n+1)}, \vec{u}^{(n)}), \tag{6}$$

where $\vec{u}^{(n)}$ denotes the state $\vec{u}$ at time $t^{(n)}$. The matrix $\overline{G}$ is an approximation to $G$ that satisfies the conditions of Theorem 1 and/or 2 required by the target system.

**Theorem 3.** *The discrete system in Eq. (6) has the discrete energy dissipation law if $\overline{G} \leq O$ and the discrete energy conservation law if $\overline{G}$ is skew-symmetric. In particular, if the system is dissipative, the amount of energy dissipation is an approximation of that of the continuous system. The system has the discrete mass conservation law if the vector $\vec{1} = (1, 1, \ldots, 1)$ is in the left kernel of $\overline{G}$.*

A discrete gradient $\overline{\nabla} H$ is not uniquely determined; hence, several methods have been proposed so far [5]. However, most methods are inapplicable to neural networks because they require a manual deformation of the system equation [15]. See Appendix A for details.

A conceptual comparison between discrete gradient methods and symplectic integrators [8, 38, 45] is summarized in Appendix E.

### 3.3 Automatic Discrete Differentiation Algorithm

To obtain a discrete gradient $\overline{\nabla} H$ of the neural networks, we propose the *automatic discrete differentiation* algorithm as an extension of the automatic differentiation algorithm [20]. Preparatorily, we introduce a discrete differential $\overline{\mathrm{d}} H$, which is a discrete counterpart of the Fréchet derivative $\mathrm{d} H$ [6];

**Definition 2.** *A discrete differential $\overline{\mathrm{d}} H : \mathbb{R}^N \times \mathbb{R}^N \times \mathbb{R}^N \to \mathbb{R}^M$ of a function $H : \mathbb{R}^N \to \mathbb{R}^M$ is a function that satisfies the following conditions;*

$$\overline{\mathrm{d}} H(a\vec{x}; \vec{u}, \vec{v}) = a\overline{\mathrm{d}} H(\vec{x}; \vec{u}, \vec{v}), \ H(\vec{u}) - H(\vec{v}) = \overline{\mathrm{d}} H(\vec{u} - \vec{v}; \vec{u}, \vec{v}), \ \overline{\mathrm{d}} H(\cdot; \vec{u}, \vec{u}) = \mathrm{d} H(\cdot; \vec{u}), \quad (7)$$

*for a scalar value $a$ and the Fréchet derivative $\mathrm{d} H(\cdot; \vec{u})$ of $H$ at $\vec{u}$.*

For a discrete differential $\overline{\mathrm{d}} H$ of a function $H : \mathbb{R}^N \to \mathbb{R}$, there exists a discrete gradient $\overline{\nabla} H$ such that $\overline{\nabla} H(\vec{v}, \vec{u}) \cdot \vec{w} = \overline{\mathrm{d}} H(\vec{w}; \vec{v}, \vec{u})$. This relationship is obvious from Definitions 1 and 2, and it is a discrete analogue of the chain-rule $\nabla H(\vec{u}) \cdot \vec{w} = \mathrm{d} H(\vec{w}; \vec{u})$.

Our proposal is to obtain a discrete differential $\overline{\mathrm{d}} H$ of the neural network model $H$ using the automatic discrete differentiation algorithm, and thereby, a discrete gradient $\overline{\nabla} H$. The automatic differentiation algorithm depends on the chain rule, product rule, and linearity. For the functions $f : \mathbb{R} \to \mathbb{R}$ and $g : \mathbb{R} \to \mathbb{R}$, it holds that

$$\frac{\partial}{\partial x}(f \circ g) = \frac{\partial f}{\partial g}\frac{\partial g}{\partial x}, \quad \frac{\partial}{\partial x}(fg) = g\frac{\partial f}{\partial x} + f\frac{\partial g}{\partial x}, \quad \frac{\partial}{\partial x}(f + g) = \frac{\partial f}{\partial x} + \frac{\partial g}{\partial x}. \tag{8}$$

**Theorem 4.** *For any $x_1, x_2, \Delta x \in \mathbb{R}$ and functions $f : \mathbb{R} \to \mathbb{R}$ and $g : \mathbb{R} \to \mathbb{R}$, the chain-rule, product rule, and linearity for the discrete differential are respectively expressed as*

$$\overline{\mathrm{d}}(f \circ g)(\Delta x; x_1, x_2) = \overline{\mathrm{d}} f(\cdot; g(x_1), g(x_2)) \circ \overline{\mathrm{d}} g(\Delta x; x_1, x_2),$$
$$\overline{\mathrm{d}}(fg)(\Delta x; x_1, x_2) = \frac{g(x_1)+g(x_2)}{2}\overline{\mathrm{d}} f(\Delta x; x_1, x_2) + \frac{f(x_1)+f(x_2)}{2}\overline{\mathrm{d}} g(\Delta x; x_1, x_2), \tag{9}$$
$$\overline{\mathrm{d}}(f + g)(\Delta x; x_1, x_2) = \overline{\mathrm{d}} f(\Delta x; x_1, x_2) + \overline{\mathrm{d}} g(\Delta x; x_1, x_2).$$

For any linear operations such as the fully-connected and convolution layers, a discrete differential is equal to the Fréchet derivative because of the linearity. For an element-wise nonlinear activation function $f : \mathbb{R} \to \mathbb{R}$, we employed the following discrete differential [18].

$$\overline{\mathrm{d}} f(\Delta x; x_1, x_2) = \begin{cases} \frac{f(x_1)-f(x_2)}{x_1-x_2}\Delta x & \text{if } x_1 \neq x_2 \\ \mathrm{d} f(\Delta x; \frac{x_1+x_2}{2}) & \text{otherwise.} \end{cases} \tag{10}$$

The product rule is applicable to bilinear operations such as attention, graph convolution, transformer, and metric function [13, 14].

Given the above, we propose the automatic discrete differentiation algorithm. With the algorithm, one can automatically obtain a discrete differential $\overline{\mathrm{d}}H$ of a neural network $H$ given two arguments, which is then converted to a discrete gradient $\overline{\nabla}H$. The computational cost is no more than twice of the ordinary automatic differentiation. The algorithm is applicable to any computational graph such as convolutional neural network [19] and graph neural network [12], and thereby one can handle extended tasks or further improve the modeling accuracy. For reference, we introduce the case with a neural network that is composed of a chain of functions in Algorithm 1 in Appendix F. We call a neural network obtaining a discrete gradient $\overline{\nabla}H$ by using the automatic discrete differentiation algorithm *DGNet*, hereafter.

### 3.4 Learning and Computation by the Discrete-Time Model

Using DGNet, we propose a deep energy-based discrete-time physical model that can learn from the discrete-time data directly as follows. Given a time series, DGNet accepts two state vectors $\vec{u}^{(n)}$ and $\vec{u}^{(n+1)}$ at time steps $n$ and $n + 1$, and then it outputs two scalar system energies $H(\vec{u}^{(n)})$ and $H(\vec{u}^{(n+1)})$. The discrete gradient $\overline{\nabla}H(\vec{u}^{(n+1)}, \vec{u}^{(n)})$ is obtained by the automatic discrete differentiation algorithm. The model is trained to minimize the squared error between the left- and right-hand sides of Eq. (6);

$$\text{minimize } \sum_n \| \tfrac{\vec{u}^{(n+1)} - \vec{u}^{(n)}}{t^{(n+1)} - t^{(n)}} - \overline{G}(\vec{u}^{(n+1)}, \vec{u}^{(n)})\overline{\nabla}H(\vec{u}^{(n+1)}, \vec{u}^{(n)}) \|_2^2 \tag{11}$$

Then, the error is back-propagated through the computational graphs including the neural network model, the discrete gradient, and the matrix $\overline{G}$ by the ordinary automatic differentiation algorithm. For training, the computational cost of the proposed scheme in Eq. (6) is no more than twice of the HNN with the Euler method and typically tens times smaller than that with the adaptive Dormand–Prince method. Through this learning process, DGNet potentially restores the true gradient $\nabla H$ from the sampled data because the discrete gradient $\overline{\nabla}H$ is equal to the true gradient $\nabla H$ when two arguments are equal by Definition 1.

For a time-series prediction, DGNet predicts the next state implicitly by solving the implicit scheme in Eq. (6) and conserves the energy strictly. The proposed discrete gradient $\overline{\nabla}H$ is time-symmetric, which implies that the proposed method is at least a second–order method [33]. Higher-order methods can be designed using the composition method (using multiple sub-steps) and the higher-order temporal difference (using multiple steps) as introduced in [17].

Indeed, the training and prediction can be performed in a different manner. After learning from the finite differences, DGNet provides the gradient $\nabla H$ so it is available for an explicit numerical method, which can be more computationally efficient and be preferable when the learned models are used in existing physics simulators (e.g., Matlab). When the true time-derivative is known, DGNet can learn it as the previous models did. Then, it can predict the next step using the discrete gradient implicitly while conserving energy.

## 4 Learning of Partial and Ordinary Differential Equations

**Comparative Models.** We examined the proposed DGNet and comparative methods. NODE is a neural network that outputs the time-derivative of the states in a general way [7]. The HNN is a neural network where the output represents the system energy $H$, and its gradient with respect to the input state $\vec{u}$ is used for the time-derivative [19]. In our experiments, they were trained from a finite difference between two successive time steps using a numerical integrator, which is similar to some previous studies [8, 38, 41, 44, 45]. For numerical integrators, we employed the explicit midpoint method (RK2) and the Dormand–Prince method with adaptive time-stepping (ada. DP); they are second– and fourth–order explicit Runge–Kutta methods. Then, the output error was back-propagated through all stages [7]. In terms of applying the HNN to the discretized PDEs, we generalized it by using the formulation in Section 3.1 and denoted it as the HNN++. DGNet was trained to minimize the objective in Eq. (11); for simplicity, the matrix $G$ of the system was assumed to be known, and we used $\overline{G} = G$. We also employed explicit numerical integrators for DGNet's prediction to reduce the computational cost from the implicit scheme in Eq. (6).

Table 2: Results on the PDE datasets.

| Model | Integrator | | KdV equation | | | Cahn–Hilliard equation | | |
| | Training | Prediction | Deriv. | Energy | Mass | Deriv. | Energy | Mass |
|---|---|---|---|---|---|---|---|---|
| NODE [7] | RK2 | RK2 | >10000 | >10000 | 2857.81 | 791.25 | >10000 | 914.72 |
| | ada.DP | ada.DP | >10000 | >10000 | 2836.45 | 790.48 | >10000 | 913.96 |
| HNN++ | RK2 | RK2 | 36.32 | 6.32 | 0.70 | 344.23 | >10000 | 87.55 |
| | ada.DP | ada.DP | 23.27 | 3.01 | 0.34 | 33.03 | 4.89 | 0.80 |
| DGNet | ↑ | RK2 | ↑ | 1.84 | 0.28 | ↑ | >10000 | 821.58 |
| | Eq. (6) | ada.DP | **17.48** | **1.60** | **0.25** | **7.14** | **0.34** | **0.07** |
| | ↓ | Eq. (6) | ↓ | **1.60** | **0.25** | ↓ | **0.34** | **0.07** |

The best and second best results are emphasized by bold and underlined fonts, respectively. Multiplied by $10^0$ for Deriv. and by $10^{-6}$ for Energy of the Cahn–Hilliard equation, and by $10^{-3}$ for the others.

**Hamiltonian PDE.** We evaluated the models on a Hamiltonian PDE, namely the KdV equation, which is a famous model that has soliton solutions [15, 16]. Of the discretized 1-dimensional KdV equation, the system energy $H$ and time evolution are expressed as follows.

$$H(\vec{u}) = \Delta x \sum_k (-\tfrac{1}{6}\alpha u_k^3 - \tfrac{1}{2}\beta(D\vec{u})_k^2), \quad \tfrac{\partial \vec{u}}{\partial t} = D\nabla H(\vec{u}) = D(-\tfrac{1}{2}\alpha(\vec{u}\odot\vec{u}) + \beta(D_2\vec{u})), \quad (12)$$

where the subscript $_k$ denotes the $k$-th element, $D$ and $D_2$ denote the first– and second–order central differences in Eq. (4), and $\odot$ denotes the element-wise multiplication. The coefficients $\alpha$ and $\beta$ determine the spatio-temporal scales. We set $\alpha = -6$, $\beta = 1$, the spatial size to 10 space units, and the space mesh size $\Delta x$ to 0.2 . At $t = 0$, we set two solitons, each of which were expressed as $-\frac{12}{\alpha}\kappa^2 \text{sech}^2(\kappa(x - d))$. $\kappa$ denotes the size randomly drawn from $\mathcal{U}(0.5, 2)$, and $d$ denotes its initial location randomly, which is determined to stay 2.0 space units away from each other. We employed the discrete gradient method in [16] to ensure the energy conservation law. We simulated the equation with a time step size of $\Delta t = 0.001$ for 500 steps and obtained 100 time series (90 for training and 10 for the test). Every experiment in this section was done with double precision.

We employed a neural network composed of a 1-dimensional convolution layer followed by two fully-connected layers. A convolution layer with a kernel size of 3 is enough to learn the central difference. The matrix $G = D$ was implemented as a 1-dimensional convolution layer with the kernel of $(-1/2\Delta x, 0, 1/2\Delta x)$ and periodic padding. Following the study on the HNN [19], the activation function was the hyperbolic tangent, the number of hidden channels was 200, and each weight matrix was initialized as a random orthogonal matrix. Each network was trained using the Adam optimizer [25] with a batch size of 200 and a learning rate of 0.001 for 10,000 iterations.

After training, we examined the mean squared error (MSE) of the time-derivative; we provided an average over 15 trials on Table 2 (see the column "Deriv."). We omitted the outliers and standard deviations for readability (see Appendix G for the full results). DGNet restored the true time-derivative well. The HNN++ employed the adaptive Dormand–Prince method, but it suffered from the gap between the time-derivative and the finite difference. Nonetheless, the application of the HNN to a PDE system is one of the contributions of this study. NODE failed to model the equation. For evaluating the long-term consistency, we predicted the test time series from the initial state $\vec{u}^{(0)}$ and obtained the MSE of the total energy and local mass (see the columns "Energy" and "Mass"). We also visualized the prediction result for each model with the best integrator, which is depicted in Fig. 2. DGNet also conserved energy the best with all integrators. Even though the implicit scheme in Eq. (6) is computationally expensive, DGNet provided the time-derivative for explicit numerical integrators, and it was enough for conserving energy in the present experiment scale

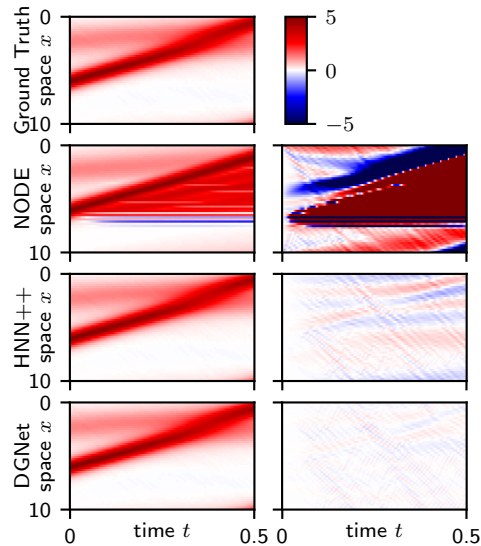

Figure 2: KdV equation. (left) Predicted state $u$. (right) Error ($\times 20$).

Table 3: Results for the ODE datasets

| Model | Integrator | | Mass-Spring | | Pendulum | | 2-Body | | Real Pendulum | |
|---|---|---|---|---|---|---|---|---|---|---|
| | Training | Prediction | Deriv. | Energy | Deriv. | Energy | Deriv. | Energy | Diff. | Energy |
| NODE | RK2 | RK2 | 52.68 | 570.32 | 56.67 | 4602.57 | 20.81 | >10000 | 1.38 | 0.62 |
| | ada.DP | ada.DP | 55.74 | 574.06 | 55.40 | 4624.66 | 20.71 | >10000 | **1.37** | 0.59 |
| HNN [19] | RK2 | RK2 | **38.22** | 61.25 | 42.49 | 404.24 | 5.39 | 93.88 | 1.42 | 2.86 |
| | ada.DP | ada.DP | 39.92 | 1.74 | 40.88 | 16.55 | 6.21 | 81.84 | 1.41 | 3.44 |
| SRNN [8] | leapfrog | leapfrog | 39.47 | 0.69 | **39.24** | 11.24 | **4.36** | **40.37** | (1.38) | (9.63) |
| DGNet | ↑ | RK2 | ↑ | 61.26 | ↑ | 743.42 | ↑ | 81.07 | ↑ | 0.86 |
| | Eq. (6) | ada.DP | 38.50 | **0.62** | 39.30 | 16.06 | 7.80 | 81.04 | 1.38 | **0.49** |
| | ↓ | Eq. (6) | ↓ | **0.62** | ↓ | **10.79** | ↓ | 81.03 | ↓ | 0.50 |

The best and second best results are emphasized by the bold and underlined fonts, respectively.
Multiplied by $10^{-6}$ for the 2-body dataset and by $10^{-3}$ for the others.

(for a longer case, see Appendix G). This result implies that the discrete gradient method provides a good framework for learning from the finite difference; to the best of our knowledge, this is the first time to confirm such contribution of the discrete gradient. In addition, one might say that the implicit scheme in Eq. (6) is as powerful as the fourth–order integrator with adaptive time stepping even though it is a second–order method.

**Dissipative PDE.** We evaluated the models on a dissipative PDE, namely the Cahn–Hilliard equation. This equation is derived from free-energy minimization and it describes, for example, the phase separation of copolymer melts [15, 16]. The system energy $H$ and time evolution of the discretized 1-dimensional Cahn–Hilliard equation are expressed as follows.

$$H(\vec{u}) = \Delta x \sum_k (\tfrac{1}{4}(u_k^2 - 1)^2 + \gamma \tfrac{1}{2}(D\vec{u})_k^2), \ \tfrac{\partial \vec{u}}{\partial t} = D_2 \nabla H(\vec{u}) = D_2((\vec{u} \odot \vec{u} - \vec{1}) \odot \vec{u} - \gamma D_2 \vec{u}), \quad (13)$$

where the coefficient $\gamma > 0$ denotes the mobility of the monomers. The mass $u_k$ has an unstable equilibrium at $u_k = 0$ (totally melted) and stable equilibria at $u_k = -1$ and $u_k = 1$ (totally separated). We set $\gamma$ to 0.0005, the spatial size to 1, the space mesh size $\Delta x$ to 0.02, the time step size $\Delta t$ to 0.0001, and the initial state $u_k$ to a random sample from $\mathcal{U}(-0.05, 0.05)$. The other conditions are the same as the case with the KdV equation.

We summarized the results in Table 2 and visualized the prediction result for each model with the best integrator in Fig. 3. DGNet outperformed the HNN++ by a large margin. The Cahn–Hilliard equation is "stiff"; this implies that the state can change drastically and an explicit integrator requires a much smaller time step size. The adaptive Dormand–Prince method evaluated the HNN++ 50–100 times per time step in the training phase and consumed the proportional computational cost. However, it did not learn the discrete-time dynamics well; the HNN++ underestimated the diffusion as shown in Fig. 3. Conversely, DGNet can estimate the dissipative term well, as expected in Theorem 3.

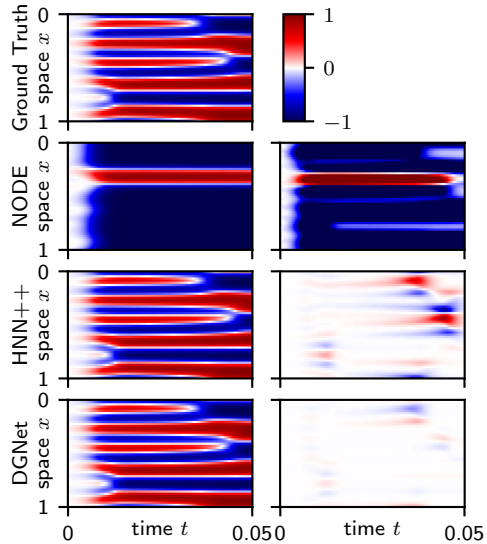

Figure 3: Cahn–Hilliard equation. (left) Predicted state $u$. (right) Error ($\times 3$).

**Hamiltonian Systems.** We employed Hamiltonian systems that were examined in the original study of the HNN [19], namely a mass-spring system, a pendulum system, and a 2-body system. Because they are natural systems, we used the matrix $G = S = \left( \begin{smallmatrix} 0 & I_n \\ -I_n & 0 \end{smallmatrix} \right)$. Instead of the time-derivative, we used the finite difference for training like the cases above. Moreover, we unified the time step size for training and test (see Appendix G for details). The other experimental settings were the same as the original experiments [19] and the cases above. Every experiment of ODEs was done with single precision. Following the symplectic recurrent neural network (SRNN) [8], we employed the leapfrog integrator and a pair of networks of the same size to represent the potential energy $V(\vec{q})$ and kinetic energy $T(\vec{p})$. The leapfrog integrator is typically applicable to this class.

We summarized the results in Table 3. DGNet sometimes obtained a worse time-derivative error but it always achieved better prediction errors than the HNN; DGNet learned the contour lines of the Hamiltonian $H$ rather than the time-derivative. DGNet achieved the best results on the long-term predictions in the mass-spring and pendulum datasets and the second-best result in the 2-body dataset. The SRNN achieved a remarkable result in the 2-body dataset because its network and integrator are specially designed for the separable Hamiltonian, which is a powerful assumption in general. DGNet for the separable Hamiltonian is a possible future study.

**Physical System with Friction.** We evaluated the models on the real pendulum dataset that were obtained by Schmidt and Lipson [39] following the study on the HNN [19]. This dataset contains the angle and angular momentum readings of a pendulum bob. Since the real pendulum has friction, we used the matrix $G = S - R = \left( \begin{smallmatrix} 0 & 1 \\ -1 & 0 \end{smallmatrix} \right) - \left( \begin{smallmatrix} 0 & 0 \\ 0 & g \end{smallmatrix} \right)$, where $g$ is an additional parameter that represents the friction and it was initialized to zero. Solved by a Runge–Kutta method, this model can be regarded as the dissipative symODEN without the control input [44].

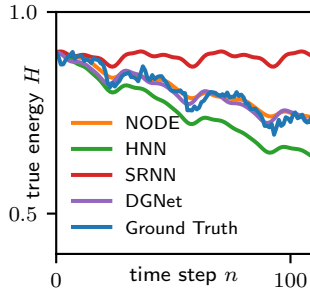

Figure 4: Results for the real pendulum dataset.

We evaluated the MSE of the finite difference (i.e., 1-step prediction, see the column "Diff.") and the MSE of the energies in long-term predictions. The results are summarized in Table 3 and Fig. 4. While all methods achieved similar errors in the 1-step prediction, the HNN achieved the worst error in the long-term prediction; the HNN overestimated the friction while DGNet estimated the friction term well, as expected in Theorem 3. The energy $H$ derived only from the angle and momentum of the pendulum bob does not monotonically decrease because the other components (e.g., the pendulum rod) are ignored. DGNet estimated the alternative energy dissipating monotonically, and it predicted the states well. NODE, which can approximate a general ODE, also worked better than the HNN. For reference, we confirmed that the SRNN failed in modeling the real pendulum dataset because of the lack of a friction term.

## 5   Conclusion

We proposed a discrete-time energy-based physical model. The approach unified and widely extended the scope of neural networks for physics simulations. Using the automatic discrete differentiation algorithm, a neural network is able to use the discrete gradient method, learn from the discrete-time data accurately, and admit the important laws of physics in discrete time.

## Broader Impact

**Novel paradigm of mathematical modeling.** For computing the physical phenomena, one has to build a difference equation in discrete time. Mathematical models for physics are typically given as differential equations, and they are discretized using numerical integrators (see the lower part of Fig. 1). This discretization may destroy the geometrical structure from which the laws of physics follow. Most previous studies on neural networks for physical phenomena employ this approach [8, 19, 38, 41, 44, 45].

The discrete gradient method is a discrete-time approximation of a continuous-time structure (see the middle part of Fig. 1) [5, 6, 15, 16, 17, 18, 24, 32, 33]. It admits the laws of physics in discrete time, but it suffers from the discretization error, too. This method has been inapplicable in neural networks until this study. We addressed this issue by introducing the automatic discrete differentiation algorithm.

Our approach is defined in discrete time and it learns discrete-time dynamics directly from discrete-time data (see the upper part of Fig. 1). As a result, it never suffers from the discretization error even though the modeling error matters. In this sense, this study provides a novel paradigm for mathematical modeling.

**Novel framework of scientific machine learning.** The proposed approach combines neural networks and geometric integration, in particular, the discrete gradient method that is derived by the

automatic discrete differentiation algorithm. As far as we know, the proposed framework is the first approach that unifies mathematical modeling from the first principles, data-driven modeling, and energetic-property-preserving numerical computations. From the viewpoint of scientific computing, the latter two may significantly accelerate scientific simulations. In practical simulations, modeling and numerical computations have been performed separately, while these must be unified because the results of the simulations often require modification of the mathematical models, and vice versa.

In addition, as implemented by PyTorch, our programming codes for the proposed framework are naturally parallelized. This implementation is the first numerical library that provides parallelized numerical simulations while using the discrete gradient method, which widely accelerates the computation in scientific simulations.

## Acknowledgments and Disclosure of Funding

Funding in direct support of this work: JST CREST Grant Number JPMJCR1914, JST PRESTO Grant Number JPMJPR16EC, JSPS KAKENHI Grant Number 19K20344, 20K11693 and schoolarship by Yoshida Scholarship Foundation.

## Footnotes

[1] https://github.com/tksmatsubara/discrete-autograd

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
