[Supplementary Material]

# Supplementary Material: Appendices

## A  Geometric Numerical Integration

Geometric numerical integration is a study on the numerical integrators of ODEs that preserve the geometric property of the target class of equations. Contrary to the classical integrators, such as the Runge–Kutta family of numerical integrators, these integrators are associated with a certain restricted target class of equations; thereby, they are designed so that important geometric properties admitted by the equations in the target class are preserved. As a consequence, the numerical solutions computed by these integrators are not only quantitatively accurate but they are also qualitatively superior to those by the classical integrators. A typical target class of the equations is the Hamilton equation, which is, in terms of geometry, defined as a symplectic-gradient flow on symplectic manifolds. The Hamilton equation is characterized by the conservation law of a symplectic form, a non-degenerate and closed 2-form (covariant 2-tensor) on the underlying symplectic manifold. Symplectic integrators are the numerical integrators that preserve this conservation law; hence, they can be in a sense considered as a discrete Hamiltonian system that is an approximation to the target Hamiltonian system. Due to this property, there should exist a corresponding Hamiltonian function, i.e., energy function, which is called a shadow Hamiltonian and it must be an approximation to the energy function of the target system. Because the shadow Hamiltonian is exactly conserved, the original energy function is conserved not exactly but highly accurately. This property of symplectic integrators is theoretically guaranteed by the existence theorem of the shadow Hamiltonian.

Because the symplectic integrators conserve only the approximation of the energy function, one may want to design energy-preserving integrators, that is, numerical integrators that preserves the energy function exactly. As such, the discrete gradient method has achieved great success. The Ge–Marsden theorem implies that there does not exist an integrator which is symplectic and energy-preserving [30]. More precisely, this theorem states that if an integrator is symplectic and energy-preserving for all Hamiltonian systems, then the integrator is completely error-free; the orbits of the numerical results must be exactly on the true orbit.

As shown above, a discrete gradient is defined in Definition 1. Because a discrete gradient is not uniquely determined from this definition, several derivation methods have been proposed. A preferred method is the average vector field method, which is second–order accurate and also conjugate-symplectic (roughly speaking, the method also approximates a symplectic method with higher order accuracy). However, most of the existing discrete gradients require explicit representation of the Hamiltonian; hence, they are not available for neural networks. An exception is the Ito–Abe method [24]

$$\overline{\nabla}_{\mathrm{IA}} H(\vec{u}, \vec{v}) = \begin{pmatrix} \frac{H((u_1,u_2,...,u_N)^\top) - H((v_1,u_2,...,u_N)^\top)}{u_1-v_1} \\ \frac{H((v_1,u_2,...,u_N)^\top) - H((v_1,v_2,...,u_N)^\top)}{u_2-v_2} \\ \vdots \\ \frac{H((v_1,v_2,...,u_N)^\top) - H((v_1,v_2,...,v_N)^\top)}{u_N-v_N} \end{pmatrix} \tag{14}$$

which requires a number of evaluations of the energy function; hence, it is computationally intractable for large systems. In contrast to the Ito–Abe discrete gradient, the discrete gradient obtained by our method is available for neural networks and computationally efficient. It requires no more than twice the computational cost of the ordinary automatic differentiation. Hence, the proposed automatic discrete differentiation algorithm is indispensable for practical application of the discrete gradient method for neural networks. See also [17, 22].

## B  The Target Equations: Geometric Ordinary and Partial Differential Equations

The target equations for this study are the differential equations with a certain geometric structure. Although the differential equations include PDEs, for simplicity, we only consider ODEs; we assume that the target PDE is semi-discretized in space so that the equation can be transformed into a system of ODEs (see Appendix D). Let $\mathcal{M}$ be a finite dimensional manifold and $T\mathcal{M}$ be the tangent bundle

of $\mathcal{M}$. A covariant 2-tensor $\omega_{\vec{u}}$ at $\vec{u} \in \mathcal{M}$ is a bi-linear continuous map $T_{\vec{u}}\mathcal{M} \times T_{\vec{u}}\mathcal{M} \to \mathbb{R}$, that is, a bi-linear continuous map that maps two tangent vectors at $\vec{u}$ into a real number. We assume that $\omega_{\vec{u}}$ is non-degenerate in the sense that for any bounded linear map $f \in T_{\vec{u}}^* \mathcal{M}$, $\omega_{\vec{u}}(\vec{v}, \cdot) = f(\cdot)$ defines a unique vector $\vec{v} \in T_{\vec{u}}\mathcal{M}$. The target class of the differential equations for this study is the equations of the following form.

$$\frac{\mathrm{d}\vec{u}}{\mathrm{d}t} = X, \quad \omega_{\vec{u}}(\vec{X}, \vec{v}) = \mathrm{d}H(\vec{v}) \quad \text{for all } \vec{v} \in T_{\vec{u}}\mathcal{M}, \tag{15}$$

where $H : \mathcal{M} \to \mathbb{R}$ is an energy function and $\mathrm{d}H$ is the Fréchet derivative of $H$. In fact, because $\omega$ is non-degenerate, the vector $\vec{X}$ in the above equation is uniquely determined. The covariant 2-tensor $\omega_{\vec{u}}$ can be written as follows.

$$\omega_{\vec{u}}(\vec{v}, \vec{w}) = \vec{w}^\top A(\vec{u}) \vec{v} \tag{16}$$

with a matrix $A(\vec{u})$; hence, Eq. (15) is shown to be equivalent to

$$\vec{w}^\top A(\vec{u}) \frac{\mathrm{d}\vec{u}}{\mathrm{d}t} = \mathrm{d}H(\vec{w}). \tag{17}$$

By using the standard inner product $\langle \cdot, \cdot \rangle$, this can be expressed as follows.

$$\left\langle A(\vec{u}) \frac{\mathrm{d}\vec{u}}{\mathrm{d}t}, \vec{w} \right\rangle = \langle \nabla H, \vec{w} \rangle, \tag{18}$$

from which it follows

$$\frac{\mathrm{d}\vec{u}}{\mathrm{d}t} = G(\vec{u}) \nabla H, \quad G(\vec{u}) = A(\vec{u})^{-1}, \tag{19}$$

where $A(\vec{u})^{-1}$ exists because $\omega_{\vec{u}}$ is non-degenerate. This is our target equation in Eq. (1).

The typical examples of the manifolds with such a 2-tensor are the Riemannian manifold [4] and the symplectic manifold [29]. In the former case, $\omega_{\vec{u}}$ is the inverse of the matrix that represents the metric tensor, which corresponds to the negative definite matrix $G(\vec{u})$. In the latter case, $\omega_{\vec{u}}$ is the symplectic form, for which the matrix $G(\vec{u})$ is skew-symmetric. They correspond to these manifolds in the target equation in Eq. (15) and they are known as the gradient flow and the symplectic flow, respectively.

## C  Proofs

This section provides the proofs of the Theorems in the main text.

*Proof of Theorem 1.*  From the chain-rule, it follows that

$$\frac{\mathrm{d}H}{\mathrm{d}t} = \nabla H \cdot \frac{\mathrm{d}\vec{u}}{\mathrm{d}t} = \nabla H^\top G \nabla H, \tag{20}$$

and this is less than or equal to 0 if $G \leq O$ and it vanishes if $G$ is skew-symmetric. $\qquad\square$

*Proof of Theorem 2.*

$$\frac{\mathrm{d}}{\mathrm{d}t} \sum_k u_k = \vec{1} \frac{\mathrm{d}u}{\mathrm{d}t} = \vec{1} G \nabla H = 0. \tag{21}$$

$\square$

*Proof of Theorem 3.*  If $G$ is negative semi-definite, it follows from the definition of the discrete gradient that

$$\frac{H(\vec{u}^{(n+1)}) - H(\vec{u}^{(n)})}{t^{(n+1)} - t^{(n)}} = \overline{\nabla} H(\vec{u}^{(n+1)}, \vec{u}^{(n)})^\top \overline{G} \overline{\nabla} H(\vec{u}^{(n+1)}, \vec{u}^{(n)}) \leq 0 \tag{22}$$

and the amount of the energy dissipation $\overline{\nabla} H(\vec{u}^{(n+1)}, \vec{u}^{(n)})^\top \overline{G} \overline{\nabla} H(\vec{u}^{(n+1)}, \vec{u}^{(n)})$ is indeed an approximation to $\nabla H(\vec{u})^\top G \nabla H(\vec{u})$, which shows the discrete energy dissipation law. Similarly, if $G$ is skew-symmetric,

$$\frac{H(\vec{u}^{(n+1)}) - H(\vec{u}^{(n)})}{t^{(n+1)} - t^{(n)}} = \overline{\nabla} H(\vec{u}^{(n+1)}, \vec{u}^{(n)})^\top \overline{G} \overline{\nabla} H(\vec{u}^{(n+1)}, \vec{u}^{(n)}) = 0 \tag{23}$$

is obtained in the same way. For the discrete mass conservation law,

$$\frac{\sum_k u_k^{(n+1)} - \sum_k u_k^{(n)}}{t^{(n+1)} - t^{(n)}} = \vec{1}\left(\frac{\vec{u}^{(n+1)} - \vec{u}^{(n)}}{t^{(n+1)} - t^{(n)}}\right) = \vec{1}\overline{G\nabla}H(\vec{u}^{(n+1)}, \vec{u}^{(n)}) = 0. \tag{24}$$

$\square$

*Proof of Theorem 4.* The first equation comes from the linearity with respect to the first argument. The third equation is obvious by definition. The second equation is a well-known result based on the studies of the discrete gradient methods [17]:

$$f(x_1)g(x_1) - f(x_2)g(x_2) = \tfrac{g(x_1)+g(x_2)}{2}(f(x_1) - f(x_2)) + \tfrac{f(x_1)+f(x_2)}{2}(g(x_1) - g(x_2)). \tag{25}$$

$\square$

# D   Semi-Discretization of the Partial Differential Equations

Although for simplicity we have only considered the ODEs in Section 3, the target equations of our approach include PDEs.

The target PDEs are equations of the following form.

$$\frac{\partial u}{\partial t} = G(u)\nabla H, \tag{26}$$

where $u$ may depend on $t$ and $x \in \mathbb{R}^n$, and $G(u)$ is a linear operator that depends on the function $u$. In the underlying functional space that admits the inner product $\langle \cdot, \cdot, \rangle$, we can consider the adjoint operator of $G(u)$ as the operator that satisfies

$$\langle Gv, w \rangle = \left\langle v, \tilde{G}w \right\rangle \tag{27}$$

for any functions $v, w$. If the space is real and finite dimensional and the inner product is the standard inner product, the adjoint operator of a matrix $G$ is $G^\top$.

Similarly to the finite dimensional equations, it follows from the relation between the Fréchet derivative and the gradient

$$\frac{\mathrm{d}H}{\mathrm{d}t} = \mathrm{d}H\left(\frac{\partial u}{\partial t}\right) = \left\langle \nabla H, \frac{\partial u}{\partial t} \right\rangle = \langle \nabla H, G\nabla H \rangle. \tag{28}$$

Thus, Eq. (26) has the energy dissipation law

$$\frac{\mathrm{d}H}{\mathrm{d}t} = \langle \nabla H, G\nabla H \rangle \leq 0 \tag{29}$$

if the operator $G(u)$ is negative semi-definite in the sense that $\langle v, Gv \rangle \leq 0$ for all $v$, and the energy conservation law

$$\frac{\mathrm{d}H}{\mathrm{d}t} = 0 \tag{30}$$

if $G(u)$ is skew-adjoint; $\tilde{G}(u) = -G(u)$.

Examples of the negative semi-definite $G$ in practical applications include

$$G = (-1)^{s-1}\frac{\partial^{2s}}{\partial x^{2s}}, \tag{31}$$

where $s$ is a non-negative integer and we define $G = -1$ for $s = 0$. This operator is negative semi-definite with respect to the $L^2$ inner product under certain boundary conditions. For example, suppose that the underlying space is the interval $[0, 1]$. Then for functions $f, g$ it holds that

$$\int_0^1 f\frac{\partial^2 g}{\partial x^2}\mathrm{d}x = -\int_0^1 \frac{\partial f}{\partial x}\frac{\partial g}{\partial x}\mathrm{d}x + \left[f\frac{\partial g}{\partial x}\right]_0^1 = \int_0^1 \frac{\partial^2 f}{\partial x^2}g\mathrm{d}x + \left[f\frac{\partial g}{\partial x}\right]_0^1 - \left[\frac{\partial f}{\partial x}g\right]_0^1$$

and hence if $f$ and $g$ satisfy

$$\left[f\frac{\partial g}{\partial x}\right]_0^1 - \left[\frac{\partial f}{\partial x}g\right]_0^1 = 0,$$

the operator $\partial^2/\partial x^2$ is negative semi-definite with respect to the $L^2$ inner product:

$$\left\langle \frac{\partial^2 f}{\partial x^2}, g \right\rangle = -\left\langle f, \frac{\partial^2 g}{\partial x^2} \right\rangle. \tag{32}$$

Similarly, the operators

$$G = \frac{\partial^{2s-1}}{\partial x^{2s-1}}, \tag{33}$$

are skew-adjoint.

In order to apply our framework to the PDEs, it would be preferable to discretize the operator $G(u)$ because the property of this operator is essential for the energy conservation or the dissipation law and the mass conservation law. If $G(u)$ is given by Eq. (31) or Eq. (33), this operator can be discretized by using the central difference operators such as Eq. (4), while preserving the desired properties. In general, suppose that the matrix $D_s$ approximates the differential operator $\partial^s/\partial x^s$ then

$$\frac{D_s + D_s^\top}{2}, \qquad \frac{D_s - D_s^\top}{2} \tag{34}$$

are in principle respectively negative/positive semi-definite and skew-adjoint approximations to $\partial^s/\partial x^s$. This design of the operator $G$ yields an approximation to the operator $G$ with the desired property and also with the desired accuracy. For further details for the structure-preserving semi-discretization, see,. e.g., [5, 17].

## E   Comparison with Symplectic Integrators

The Ge–Marsden theorem shows that no method can be both symplectic and strictly energy-preserving [30]. Hence, the proposed discrete gradient method does not conflict with but complements a neural network model solved by a symplectic integrator [8, 38, 45]. One can choose a preferable one depending on targeted tasks.

Some symplectic integrators (such as the variational integrator) are known to preserve the momentum in a physical system. The proposed discrete gradient method can have the property of conjugate symplecticity, which guarantees the preservation of the momentum with high accuracy [21]; in fact for a certain class of problems the discrete gradient derived by the proposed algorithm is equivalent to the average vector field method, which is known to be conjugate symplectic of order four [5]. Moreover, if a certain quantity other than the energy should be strictly preserved, one can design another discrete gradient by the method proposed in [11] so that both the energy and the quantity are preserved.

With varying time-step, symplectic integrators are in general known to lose symplecticity and cannot preserve the system energy just like non-symplectic Runge–Kutta methods [22], while the proposed discrete gradient method can. In particular, depending on the learned parameters, the proposed method can choose the time-step while preserving the energy.

## F   Automatic Discrete Differentiation Algorithm

Many practical implementations of the automatic differentiation algorithm indeed obtain gradients directly [31]; hence, they are sometimes called the autograd algorithm. A discrete version of the autograd algorithm is enough for this study. From this viewpoint, the implementation of the discrete autograd algorithm is introduced as follows.

The Fréchet derivative $\mathrm{d}g(\cdot; \vec{u}) : \mathbb{R}^N \to \mathbb{R}^M$ of a function $g : \mathbb{R}^N \to \mathbb{R}^M$ at $\vec{u}$ is a bounded linear operator that satisfies the following condition.

$$\lim_{||\vec{h}|| \to +0} \frac{||g(\vec{u} + \vec{h}) - g(\vec{u}) - \mathrm{d}g(\vec{h}; \vec{u})||}{||\vec{h}||} = 0. \tag{35}$$

Figure A1: A conceptual comparison between the autograd algorithm (left) and the discrete autograd algorithm (right).

---

**Algorithm 1** Discrete Autograd Algorithm

---

**Input:** a function $H(\cdot) = f_N \circ \cdots \circ f_1(\cdot)$ for $i \in \{1, \ldots, N\}$, and arguments $\vec{u}$ and $\vec{v}$
**Output:** discrete gradient $\overline{\nabla} H(\vec{u}, \vec{v})$

$\vec{h}_0 \leftarrow \vec{u}, \vec{k}_0 \leftarrow \vec{v}$
**for** $i = 1, \cdots, N$ **do**
    $\vec{h}_i \leftarrow f_i(\vec{h}_{i-1}), \vec{k}_i \leftarrow f_i(\vec{k}_{i-1})$
**end for**
$H(\vec{u}) \leftarrow \vec{h}_N, H(\vec{v}) \leftarrow \vec{k}_N$
$\overline{\nabla} H(\vec{h}_N, \vec{k}_N) \leftarrow \mathbb{1}$
**for** $i = N, \cdots, 1$ **do**
    **if** $f_i$ is a linear layer **then**
        $\bar{J}_{f_i} \leftarrow J_{f_i}$
    **else if** $f_i$ is an element-wise nonlinear activation function **then**
        $\bar{J}_{f_i} \leftarrow \mathrm{diag}(\vec{h}_i - \vec{k}_i)\mathrm{diag}(\vec{h}_{i-1} - \vec{k}_{i-1})^{-1}$
    **else**
        $\bar{J}_{f_i} \leftarrow$ (depending on the function)
    **end if**
    $\overline{\nabla} H(\vec{h}_{i-1}, \vec{k}_{i-1}) \leftarrow \bar{J}_{f_i}^\top \overline{\nabla} H(\vec{h}_i, \vec{k}_i)$
**end for**
**return** $\overline{\nabla} H(\vec{u}, \vec{v}) \leftarrow \overline{\nabla} H(\vec{h}_0, \vec{k}_0)$

---

The Fréchet derivative $\mathrm{d}g$ can be regarded as multiplication by the Jacobian matrix $J_g(\vec{u}) \in \mathbb{R}^{M \times N}$ at $\vec{u}$.

$$\mathrm{d}g(\vec{w}; \vec{u}) = J_g(\vec{u})\vec{w}. \tag{36}$$

The chain-rule can be rewritten as a chain of Jacobian matrices.

$$\mathrm{d}(f \circ g)(\vec{w}; \vec{u}) = \mathrm{d}f(\cdot; g(\vec{u})) \circ \mathrm{d}g(\vec{w}; \vec{u}) = J_f(g(\vec{u}))J_g(\vec{u})\vec{w}. \tag{37}$$

The gradient $\nabla f$ of a scalar-valued function $f : \mathbb{R}^M \to \mathbb{R}$ is defined using an inner product $\cdot$ as follows.

$$\lim_{||\vec{k}|| \to +0} \frac{||f(\vec{v} + \vec{k}) - f(\vec{v}) - \nabla f(\vec{v}) \cdot \vec{k}||}{||\vec{k}||} = 0. \tag{38}$$

This implies that the gradient is dual to the derivative. Hence, the gradient $\nabla f$ is equal to the transposed Jacobian matrix $J_f$.

$$\mathrm{d}f(\vec{w}; \vec{v}) = J_f(\vec{v})\vec{w} = \nabla f(\vec{v}) \cdot \vec{w}. \tag{39}$$

Therefore, the gradient $\nabla(f \circ g)$ of the compositional function $f \circ g$ is obtained by multiplying the upper-layer gradient $\nabla f$ by the transposed lower-layer Jacobian matrix $J_g(\vec{u})^\top$.

$$\nabla(f \circ g)(\vec{u}) = J_g(\vec{u})^\top \nabla f(g(\vec{u})). \tag{40}$$

This is the autograd algorithm to obtain the gradient of a compositional function, which is shown in the left panel of Fig. A1.

For the discrete autograd algorithm, we replace the Jacobian matrices $J_g$ with their discrete counterparts $\bar{J}_g$ as shown in the right panel of Fig. A1. For a linear layer $g$, the discrete Jacobian matrix $\bar{J}_g$ is

equal to the ordinary one $J_g(\vec{v}) = \frac{\partial g}{\partial \vec{v}}$. For an element-wise nonlinear activation layer $g$, the discrete Jacobian matrix $\bar{J}_g$ is a diagonal matrix where each non-zero element is expressed as $\frac{f(v_1) - f(v_2)}{v_1 - v_2}$ when given two scalar arguments $v_1$ and $v_2$. If the two arguments $v_1$ and $v_2$ are closer than $\epsilon$, we use the gradient $\frac{df}{dz}$ at the midpoint $z = \frac{v_1 + v_2}{2}$ to avoid the loss of significance. We empirically found that $\epsilon = 10^{-6}$ and $\epsilon = 10^{-12}$ worked well with single and double precisions, respectively. We summarize the discrete autograd algorithm in Algorithm 1.

For obtaining a discrete gradient, the automatic discrete differentiation algorithm requires two forward paths and one modified backward path. The computational cost of a discrete gradient is one and a half times as much as that of the ordinary gradient. The training of DGNet is less expensive than the training of HNN with the explicit midpoint method (RK2), and it is tens of times less expensive than the training of HNN with the adaptive Dormand–Prince method.

# G  Details of Datasets and Results

We implemented all codes using Python v3.7.3 with libraries; numpy v1.16.2, scipy v1.2.1, and PyTorch v1.4.0 [31]. We performed all experiments on NVIDIA TITAN V for PDEs and GeForce 2080 Ti for ODEs. We also used torchdiffeq v0.0.1 library for numerical integrations of neural network models [7]. The results are summarized in Tables A1, A2, and A3.

**PDE systems.** We provide detailed results on Tables A1, which corresponds to Table 2. The standard deviations are preceded by plus-minus signs $\pm$, and the scales are noted in brackets. For all errors, the standard deviations are large. The performance difference between HNN++ and DGNet would not be significant for the KdV equation but it is obviously significant for the Cahn–Hilliard equation.

We also provide the results of longer-time predictions of the KdV equation in Fig. A2. Under the conditions same as in Section 4, HNN++ and DGNet predicted state $u$ for 5,000 steps, as shown in the left column. In the center column, the true equation in Eq. (12) gives the system energies of the true state and predicted state, as denoted by the blue and orange lines, respectively. The system energy of the true state is conserved within a range of the rounding error. The prediction errors become larger when two solitons collide with each other, but they are restored to their former levels; each model learned collisions qualitatively rather than quantitatively. In the right column, each panel shows the system energy of the predicted state when a trained neural network model is considered as a true equation. The neural network models formed Hamiltonian systems in Eq. (1) and may conserve the system energy. However, the system energy learned by HNN++ increases, implying that the conservation law is destroyed. This phenomena is called *energy drift* and occurs commonly in a Runge–Kutta method integrating a Hamiltonian system. The energy drift is a practical issue for a simulation of molecular dynamics and solar systems, where the number of time steps is more than one million and the numerical error becomes more significant than the modeling error. This is the main reason why structure-preserving integrators are needed [22]. The Runge–Kutta method also destroys the conservation law that DGNet potentially produces (see the third row). Only when using the implicit scheme in Eq. (6), DGNet gives the system energy that fluctuates within a range of $\pm 3 \times 10^{-12}$, implying that DGNet conserves the system energy only with the rounding error.

**ODE systems.** In Section 4, we evaluated the models on the ODE datasets from the original source codes for the study on the HNN [19]. Each of their datasets is composed of three parts; the first is for training, the second is for evaluating the time-derivative error, and the third is for evaluating the accuracy of the long-term prediction. The number of observations and the duration of each trajectory are summarized in Table A4. For the pendulum dataset, each trajectory in the training and test sets consists of 45 observations over three unit times. Meanwhile for the long-term prediction set, it was composed of 100 observations over 20 unit times. This difference did not matter in the original study because each model was trained with the true time-derivative and it used the adaptive Dormand–Prince method for the time-series prediction. Conversely, in our experiments, the finite difference was instead given. The difference in the time step size caused unanticipated impacts on all models. Hence, in Section 4, we set the conditions for training and testing to the same as those for the long-term prediction. Moreover, for the spring dataset prediction, the time step size was rescaled to compensate for the observation noise in the original implementation. However, we found that this modification did not matter in our experiments; thus, we removed the rescaling.

Figure A2: Results of long-term predictions. Models and integrators are on the left edge. (a) State $u$. (b) The system energies of the true state and predicted state derived from the true equation, denoted by blue and orange lines, respectively. (c) The system energy of the predicted state when considering the trained neural network as a true equation.

We provide detailed results on Tables A2 and A3, which correspond to Table 3. The standard deviations are preceded by plus-minus signs $\pm$, and the scales are noted in brackets. For the real pendulum dataset, the long-term prediction error of DGNet is significantly smaller than that of the HNN; DGNet extimates the amount of energy dissipation well, as expected in Theorem 3. Even through other differences between the HNN and DGNet would not be significant, this fact is still remarkable. For the HNN, the Dormand–Prince method is a fourth–order method and adjusts the time step size to suppress the prediction error smaller than a given threshold; it is the most numerically accurate and realible integrator in our experiments. Conversely, DGNet and the leapfrog integrator are second–order methods. For learning physical phenomena, the qualitative property is important equally to or more than the quantitative accuracy.

We provide the detailed results of the mass-spring dataset in Fig. A3. The leftmost column shows trajectories of state $\vec{u}$, each of which forms a circle. In the second left column, the true equation gives the system energy of the true state and predicted state. Because the ground truth data was generated using a Runge–Kutta method (specifically, the adaptive Dormand–Prince method implemented in solve_imp method of scipy library), the true system energy is drifting. For all neural network models, the system energy was fluctuating over a wide range due to the modeling error. Each of

Figure A3: Detailed results of the mass-spring dataset. Models and integrators are on the left edge. (a) A trajectory of state $u$. (b) The system energies of the true state and predicted state derived from the true equation, denoted by blue and orange lines, respectively. (c)(d) The system energy of the predicted state when considering the trained neural network as a true equation (c) for 100 steps, and (d) for 10,000 steps.

the remaining panels shows the system energy of the predicted state when a trained neural network model is considered as a true equation for 100 steps in the second right column and 10,000 steps in the rightmost column. Using the Runge–Kutta method (specifically, the adaptive Dormand–Prince method implemented in odeint method of torchdiffeq library), the system energies of HNN++ and

Figure A4: Detailed results of the real pendulum dataset. Models and integrators are on the left edge. (a) A trajectory of state $u$. (b) The system energies of the true state and predicted state derived from the true equation, denoted by blue and orange lines, respectively. (c) The system energy of the predicted state when considering the trained neural network as a true equation.

DGNet are drifting. Using the leapfrog integrator, the system energy of SRNN is fluctuating over the widest range. The leapfrog integrator is a symplectic integrator and conserves the "shadow" Hamiltonian, which is an approximation to the true Hamiltonian [22]. The fluctuation makes it difficult to evaluate the energy efficiency of a system, and this is a main drawback of symplectic

integrators. Using the implicit scheme in Eq. (6), the system energy of DGNet fluctuates within the narrowest range, demonstrating the superiority of the discrete gradient method.

We provide the detailed results of the real rendulum dataset in Fig. A4. The left column shows trajectories of state $\vec{u}$. Each trajectory forms a spiral except for the leapfrog integrator, which produces a circle without dissipation. In the center column, the true equation gives the system energy of the true state and predicted state. The true energy derived only from the angle and momentum of the pendulum bob does not monotonically decrease because it ignores the other components (e.g., the pendulum rod). The right column shows the system energy of the predicted state when a trained neural network model is considered as a true equation. The system energy estimated by HNN and DGNet dissipates monotonically (see green lines); these models build the alternative energy by implicitly estimating the other components. For a real-world problem, we cannot always observe all states, and then, the "true" equation cannot describes the time evolution. The data-driven modeling enables us to predict a partially observable state with the conservation and dissipation laws.

Table A1: Detailed results on the PDE datasets corresponding to Table 2.

| Model | Integrator | | KdV equation | | | Cahn–Hilliard equation | | |
|---|---|---|---|---|---|---|---|---|
| | Training | Prediction | Deriv. | Energy | Mass | Deriv. | Energy | Mass |
| NODE [7] | RK2 | RK2 | $1.15 \pm 0.01$ ($\times 10^1$) | $4.57 \pm 3.62$ ($\times 10^4$) | $2.86 \pm 0.38$ ($\times 10^0$) | $7.91 \pm 0.03$ ($\times 10^2$) | $1.43 \pm 0.05$ ($\times 10^{-2}$) | $9.15 \pm 0.11$ ($\times 10^{-1}$) |
| | ada.DP | ada.DP | $1.15 \pm 0.01$ ($\times 10^1$) | $5.58 \pm 6.56$ ($\times 10^4$) | $2.84 \pm 0.43$ ($\times 10^0$) | $7.90 \pm 0.03$ ($\times 10^2$) | $1.42 \pm 0.05$ ($\times 10^{-2}$) | $9.14 \pm 0.11$ ($\times 10^{-1}$) |
| HNN++* | RK2 | RK2 | $3.63 \pm 4.14$ ($\times 10^{-2}$) | $6.32 \pm 10.26$ ($\times 10^{-3}$) | $7.00 \pm 10.80$ ($\times 10^{-4}$) | $3.44 \pm 0.20$ ($\times 10^2$) | $1.33 \pm 1.40$ ($\times 10^{-1}$) | $8.76 \pm 5.07$ ($\times 10^{-2}$) |
| | ada.DP | ada.DP | $2.33 \pm 2.19$ ($\times 10^{-2}$) | $3.01 \pm 4.81$ ($\times 10^{-3}$) | $3.35 \pm 4.92$ ($\times 10^{-4}$) | $3.30 \pm 0.75$ ($\times 10^1$) | $4.89 \pm 1.74$ ($\times 10^{-6}$) | $7.95 \pm 3.12$ ($\times 10^{-4}$) |
| DGNet | ↑ | RK2 | ↑ | $1.84 \pm 1.69$ ($\times 10^{-3}$) | $2.78 \pm 2.77$ ($\times 10^{-4}$) | ↑ | $6.61 \pm 3.34$ ($\times 10^0$) | $8.22 \pm 2.04$ ($\times 10^{-1}$) |
| | Eq. (6) | ada.DP | $1.75 \pm 1.00$ ($\times 10^{-2}$) | $1.60 \pm 1.65$ ($\times 10^{-3}$) | $2.54 \pm 2.91$ ($\times 10^{-4}$) | $7.14 \pm 8.27$ ($\times 10^0$) | $3.39 \pm 4.05$ ($\times 10^{-7}$) | $6.95 \pm 7.64$ ($\times 10^{-5}$) |
| | ↓ | Eq. (6) | ↓ | $1.60 \pm 1.65$ ($\times 10^{-3}$) | $2.55 \pm 3.00$ ($\times 10^{-4}$) | ↓ | $3.40 \pm 4.10$ ($\times 10^{-7}$) | $6.96 \pm 7.70$ ($\times 10^{-5}$) |

Table A2: Detailed results on the ODE datasets corresponding to Table 3.

| Model | Integrator | | Mass–Spring | | Pendulum | | 2-Body | |
|---|---|---|---|---|---|---|---|---|
| | Training | Prediction | Deriv. | Energy | Deriv. | Energy | Deriv. | Energy |
| NODE | RK2 | RK2 | $5.27 \pm 0.32$ ($\times 10^{-2}$) | $5.70 \pm 1.53$ ($\times 10^{-1}$) | $5.67 \pm 0.56$ ($\times 10^{-2}$) | $4.60 \pm 0.76$ ($\times 10^0$) | $2.08 \pm 0.40$ ($\times 10^{-5}$) | $1.44 \pm 1.96$ ($\times 10^{-1}$) |
| | ada.DP | ada.DP | $5.57 \pm 0.38$ ($\times 10^{-2}$) | $5.74 \pm 1.51$ ($\times 10^{-1}$) | $5.54 \pm 0.61$ ($\times 10^{-2}$) | $4.62 \pm 0.76$ ($\times 10^0$) | $2.07 \pm 0.41$ ($\times 10^{-5}$) | $1.83 \pm 3.05$ ($\times 10^{-1}$) |
| HNN [19] | RK2 | RK2 | $3.82 \pm 0.09$ ($\times 10^{-2}$) | $6.13 \pm 1.50$ ($\times 10^{-2}$) | $4.25 \pm 0.24$ ($\times 10^{-2}$) | $4.04 \pm 0.69$ ($\times 10^{-1}$) | $5.39 \pm 2.65$ ($\times 10^{-6}$) | $9.39 \pm 8.10$ ($\times 10^{-5}$) |
| | ada.DP | ada.DP | $3.99 \pm 0.09$ ($\times 10^{-2}$) | $1.74 \pm 3.99$ ($\times 10^{-3}$) | $4.09 \pm 0.29$ ($\times 10^{-2}$) | $1.66 \pm 0.59$ ($\times 10^{-2}$) | $6.21 \pm 4.65$ ($\times 10^{-6}$) | $8.18 \pm 6.08$ ($\times 10^{-5}$) |
| SRNN [8] | leapfrog | leapfrog | $3.95 \pm 0.08$ ($\times 10^{-2}$) | $6.90 \pm 8.08$ ($\times 10^{-4}$) | $3.92 \pm 0.14$ ($\times 10^{-2}$) | $1.12 \pm 0.67$ ($\times 10^{-2}$) | $4.36 \pm 2.40$ ($\times 10^{-6}$) | $4.04 \pm 3.22$ ($\times 10^{-5}$) |
| DGNet | ↑ | RK2 | ↑ | $6.13 \pm 0.74$ ($\times 10^{-2}$) | ↑ | $7.43 \pm 1.23$ ($\times 10^{-1}$) | ↑ | $8.11 \pm 10.91$ ($\times 10^{-5}$) |
| | Eq. (6) | ada.DP | $3.85 \pm 0.09$ ($\times 10^{-2}$) | $6.16 \pm 4.28$ ($\times 10^{-4}$) | $3.93 \pm 0.19$ ($\times 10^{-2}$) | $1.61 \pm 1.16$ ($\times 10^{-2}$) | $7.80 \pm 4.22$ ($\times 10^{-6}$) | $8.10 \pm 10.91$ ($\times 10^{-5}$) |
| | ↓ | Eq. (6) | ↓ | $6.17 \pm 4.28$ ($\times 10^{-4}$) | ↓ | $1.08 \pm 0.91$ ($\times 10^{-2}$) | ↓ | $8.10 \pm 10.91$ ($\times 10^{-5}$) |

Table A3: Detailed results on the real pendulum dataset corresponding to Table 3.

| Model | Integrator | | Real Pendulum | |
| | Training | Prediction | Deriv. | Energy |
|-------|----------|-----------|--------|--------|
| NODE | RK2 | RK2 | $1.38 \pm_{0.02}$ ($\times 10^{-3}$) | $6.22 \pm_{4.32}$ ($\times 10^{-4}$) |
| | ada.DP | ada.DP | $1.37 \pm_{0.02}$ ($\times 10^{-3}$) | $5.88 \pm_{4.02}$ ($\times 10^{-4}$) |
| HNN [19] | RK2 | RK2 | $1.42 \pm_{0.22}$ ($\times 10^{-3}$) | $2.86 \pm_{0.50}$ ($\times 10^{-3}$) |
| | ada.DP | ada.DP | $1.41 \pm_{0.15}$ ($\times 10^{-3}$) | $3.44 \pm_{1.71}$ ($\times 10^{-3}$) |
| SRNN [8] | leapfrog | leapfrog | $1.38 \pm_{0.02}$ ($\times 10^{-3}$) | $9.63 \pm_{0.49}$ ($\times 10^{-3}$) |
| | $\uparrow$ | RK2 | $\uparrow$ | $8.63 \pm_{4.96}$ ($\times 10^{-4}$) |
| DGNet | Eq. (6) | ada.DP | $1.38 \pm_{0.06}$ ($\times 10^{-3}$) | $4.92 \pm_{3.94}$ ($\times 10^{-4}$) |
| | $\downarrow$ | Eq. (6) | $\downarrow$ | $5.04 \pm_{3.99}$ ($\times 10^{-4}$) |

Table A4: Details of the datasets used in [19]

| Dataset | #Iteration | Training/Test | | | Long-Term Prediction | | |
| | | #Traject. | #Observ. | Duration | #Traject. | #Observ. | Duration |
|---------|-----------|-----------|----------|----------|-----------|----------|----------|
| Mass-Spring | 2,000 | 25/25 | 30 | 3 | 15 | 100 | 20 |
| Pendulum | 2,000 | 25/25 | 45 | 3 | 15 | 100 | 20 |
| 2-Body | 10,000 | 800/200 | 50 | 20 | 15 | 500 | 25 |