[Reviews · NeurIPS 2020]

Review 1

Summary and Contributions: This article proposes learning physical energy-based physical models from discrete time data using neural networks by explicitly embedding conservation or dissipation laws of energy in the system. This is achieved by parametrising the system energy of an energy-based dynamical system with a neural network. In order to ensure conservation or dissipation of energy of the equation's discrete form, they propose using a discrete gradient, which by construction conserves the underlying continuous system's energy, as opposed to previous works. This has the advantage of being able to learn the energy function with discrete time data, and not suffering from discretisation errors introduced in the learning phase. The approach is evaluated on examples from different classes of dynamical systems (dissipative PDEs, Hamiltonian PDEs).

Strengths: Data-driven modelling of dynamical systems is clearly an important and actual field of research. The article is rigorous and well written. The proposed method generalises previous approaches, attacking a more general family of dynamical systems as previous works in a novel and interesting way. As the proposed method does not require time-integration during the learning phase, this method could require little smaller time and memory complexity, than back-propagating the errors from multiple time-steps.

Weaknesses: It seems that once the model is learned from discrete data, an explicit numerical scheme is used in the prediction phase. Does this mean that the system's energy is no longer conserved/dissipated as in the continuous dynamical system?

Correctness: The experimental seems to be correct, and to have been done conscientiously.

Clarity: Yes, the paper is very well written.

Relation to Prior Work: Yes

Reproducibility: Yes

Additional Feedback: It would have been interesting too see analyse the results of the different methods by artificially varying the time-step between consecutive data points: how does this affect training and generalisation for both the estimation of the energy and the predictions ? Could the temporal discretisation be chosen after training, based on the learned parameters?


Review 2

Summary and Contributions: There has been a lot of recent work in the broad area of 'Hamiltonian Neural Nets' - but all have worked on continuous functions, including continuous time, with the tacit assumption that samples observed are discretizations (samples) from these systems. None have looked explicitly at how such methods should and could be applied directly to discrete data, such as obtained by finite grid solutions, or delay-difference models. The paper considers the discrete systems, developing sample-based alternatives to standard operators and proposes a generic discrete gradient method. The paper showcases the approaches on a variety of problems, including some 'real world'. The method claims to be more general as it can learn from PDEs as well as ODEs, and it avoids discretisation errors that arise from using RK integrators. The method does this by a) introducing a general formalism which highlights how the rate of change of states can be cast into a matrix G times the gradient of a Hamiltonian and b) by discretising the gradient of the Hamiltonian with a Frechet derivative.

Strengths: The approach is sound, and I checked all the relevant math (to best of my ability). The approach is shown to work well empirically in a variety of problems. The delta with prior work is detailed by the authors - highlighting the novelty of the contributions made, primarily in the breadth of applicability and the discrete auto-grad. There is a lot of interest in the generic HNN methods, so this will be of relevance to the NeurIPS community. Pros of the paper include: * Can solve a broader class of problems compared to current solutions in the field by modeling an additional G matrix e.g. Friction systems (ODE and PDE) & Discrete PDEs (PDE) * Maintains energy and mass conservation via discretisation which preserves the geometric structure (a.k.a volume preservation) * Good form for discrete auto differentiation * Good PDE dataset motivations * Can extend research to identify discrete PDEs

Weaknesses: The paper has a few areas that could be improved, and some conceptual questions arise * there is no mention of momentum conservation (I know you discuss mass conservation) * There is no study (at least not explicit) of higher-order systems e.g. all the data descriptions talk about single precision (does this mean Euler/RK2??) what happens if we have data that has/requires higher precision? * There is not much novelty in the math - the use of the G matrix is mentioned in other papers for example * ODE systems are not appropriately benchmarked - though maybe this is an issue with paper length? Dissipative systems are talked about but SympODEN dissipative is not benchmarked (although dissipative SympODEN is technically new and not required to be benchmarked perhaps?) * Precision is talked about in Hamiltonian systems but VIN isn’t benchmarked * The method appears not readily extendable to N particle systems, unlike in Graph Hamiltonian approaches (VIN and extensions like VIGN). * The computational overhead seems prohibitive (2x slower than HNN which could suffer in large data/state space domains - again this is where VIN style approaches can win big) * A mention of how control might be incorporated via G would be interesting

Correctness: The paper is (from my checking) technically correct. The theory, methods development and empirical methodology all are well structure and presented.

Clarity: Mainly well written - though on occasion the sentence structure is tough to pick apart (probably due to 'compression' for the submission). Still, I enjoyed reading this paper.

Relation to Prior Work: Yes - mainly. There is sparse commentary of work like VIN and its recent extensions, which would seem better suited to discrete methods than HNN (any comments on this?). I know they are not directly related to this work, but there are a lot of papers on ODE and PDE solutions (including discrete cases) in the 'probabilistic' Gaussian Process literature. the briefest of mentions would help the reader get a fuller picture of the domain.

Reproducibility: Yes

Additional Feedback: Main high level comments as above - a few extra thoughts / edits / comments below: * recapping my comment on Lagrange / VIN / VIGN - which naively would seem even better suited to the approaches you take. You don't mention these (much) - any reasons? * p2. you define here \vec{q} to be the potential, and later the generalized location. the two _can_ be the same, but not generally. * Just above Eq 1, you define H as a map from R^N to R before making explicit that H is H(u) and u \in R^N. * Thm. 1 G \leq 0. I understand the abuse of notation, but make clear at first usage that you mean the spectral components of G (ie the eigen norm) \leq 0. Relate to eg Lyapunov exponents for clarity? You use this shorthand several times (like S-R etc) * Rmk. 1 reference for Cahn–Hilliard and KdV equations? Some will find this a bit confusion as KdV is often discussed as a continuous time PDE. Worth a footnote to comment on usage of discrete versions of these and practical importance? * Burgers eqn -> Burgers' eqn + give cite * Eq. 8 - what is the 'o' operator - define * "κ denotes the size randomly drawn from U (0.5, 2), and d denotes its initial location randomly, which is determined to stay 2.0 away from each other. " : sentence needs rewriting. Also, what exactly do you mean 'stay 2.0 from each other'? Dear authors. Thank you for your response to my comments, and those of the other reviewers. I still believe there is merit in this work, though it still could benefit from some alterations and extensions (including potential comparative studies). This work is, in my opinion, still marginally above the acceptance threshold.


Review 3

Summary and Contributions: The paper concerns neural network based learning of Hamiltonian systems. A novel approach involving discrete gradients is formulated, argued for and placed in relation to earlier approaches. The authors have implemented an automatic discrete gradient framework (in the spirit of autodiff but for discrete gradients). The methods is tested on data simulated from several Hamiltonian systems.

Strengths: This paper is in my view excellent: - the idea of using discrete gradients is novel and, as far as I can judge, very well-suited for the problem context (learning Hamiltonian systems) - the exposition and the mathematical presentation (including the general descriptions of the systems in section 3.1) are precise and clear - the implementation of automatic discrete differentiation presents an new take on autodiff and backpropagation - the framework applies to ODEs and PDEs - the method is shown to perform well in practice

Weaknesses: None

Correctness: The paper is to the best of my judgement correct

Clarity: The paper is well written and the exposition clear.

Relation to Prior Work: Yes

Reproducibility: Yes

Additional Feedback: This is definitely the most interesting paper in my batch.

[Author Response · NeurIPS 2020]



**an explicit numerical scheme is used in the prediction phase:** In the prediction phase, we have employed both the proposed discrete gradient (see "Eq. (6)" in Tables 2-3 in Section 4 and Figs. A2–A4 in Appendix F) and explicit numerical schemes (see also RK and ada.DP). With the discrete gradient method, the system's energy is strictly conserved/dissipated, and with the explicit numerical schemes, as you indicate, it is not; however, the latter would be preferable when the learned models are used in existing physics simulators (e.g., Matlab). One can choose either scheme depending on the situations, and this is one of the advantages of our proposed method.

**artificially varying the time-step:** We appreciate the very interesting comments. With varying time-step, symplectic integrators (the leapfrog integrator in our study) are known to lose symplecticity and cannot preserve the system energy just like non-symplectic Runge–Kutta methods, while the proposed discrete gradient method can. In particular, depending on the learned parameters, the proposed method can choose the time-step while preserving the energy.

**Replies to Reviewer #3.**

**momentum conservation:** A discrete gradient method can have the property of conjugate symplecticity, which guarantees the preservation of the momentum with high accuracy (E. Hairer, *Journal of Numerical Analysis*, 2010). Moreover, if a certain quantity should be strictly preserved, one can design a modified discrete gradient by a method proposed by M. Dahlby et al. *Journal of Physics A*, 2011, so that both the energy and the quantity are preserved. We will discuss these potential extensions as future work in the final version.

**higher-order systems:** Single precision (32-bit floating-point numbers) was used for ODE datasets, and double precision (64-bit) for PDE, as stated in lines 236 and 298. Although, as stated in line 205, the proposed discrete gradient method is a second-order method, higher-order methods can be designed using the composition method (using multiple sub-steps) and the higher-order temporal difference (using multiple steps) as introduced in [12]. We will add the above explanation in the final version. Moreover, as shown in Tables 2–3, the proposed discrete gradient method works with a smaller error than the Dormand–Prince method with the adaptive time-stepping, which is a fourth-order method.

**novelty in the math:** Firstly, the automatic discrete differentiation algorithm is a novel mathematical idea. Secondly, dissipative SymODEN in [35] introduced a matrix similar to $G$ in our paper as the standard skew-symmetric matrix minus a symmetric positive semi-definite matrix. From a geometric point of view, this means that the systems are defined on cotangent bundles, while our approach is formulated on general symplectic/Riemannian manifolds. This extension enables our method to handle the PDE systems (e.g., the Cahn–Hilliard equation and the KdV equation).

**Lagrange / VIN / VIGN, particularly, VIN isn't benchmarked:** Our study focuses on both energy-conservative and -dissipative systems. To this end, the Hamiltonian formalism is suitable because it has the equation of the same form as the dissipative gradient flows on Riemannian manifolds. In fact, it can express many kinds of conservative and dissipative ODEs and PDEs in a single unified way as shown in Eq. (1) in our paper, emphasizing the impact of the proposed discrete gradient. Conversely, Lagrangian mechanics is not obviously extendable to general dissipative systems (e.g., the Cahn–Hilliard equation), and the variational integrator used by VIN is a second–order symplectic integrator derived from Lagrangian mechanics. Instead, we have evaluated the leapfrog integrator, which is also a second–order symplectic integrator and is applicable to Hamiltonian formalism. We will add these explanations in the final version along with the suitable references for the Lagrangian approaches.

**SympODEN dissipative is not benchmarked:** SympODEN dissipative employed a dissipative term (in the same way as Eq. (3) in our paper) and a control input (not needed in our experiments), and learned from finite difference using a Runge–Kutta method (in the same way as our experiments). Hence, all proposals of SympODEN dissipative were already considered or out of scope. In the final version, we will explain this fact clearly.

**extendable to N particle systems:** Our proposed automatic discrete differentiation algorithm is defined using the Fréchet derivative, and thus it is applicable to any kind of network architectures. This means that the proposed DGNet is extendable to operations on a graph and thereby to N particle systems in the same way as VIN is extended to VIGN.

**The computational overhead:** For training, the computational cost of the proposed scheme in Eq. (6) is no more than twice of the HNN with the Euler method and tens times smaller than that with the adaptive Dormand–Prince method. Besides, the Ge–Marsden theorem shows that no method can be both symplectic and strictly energy-preserving (see G. Zhong and J. Marsden, Physics Letters A, 1988). Hence DGNet does not conflict with but complements VIN. One can choose a preferable one depending on targeted tasks.

**control might be incorporated:** In the same way as dissipative SymODEN, DGNet can be extended to a port-Hamiltonian system that accepts the control input. We will add some comments on this in the final version.

**Additional:** We thank you for the comments on our bobtailed explanations. We will clarify them in the final version.

**Reply to Reviewer #4:** We deeply thank you for your review and appreciation.

[Meta-Review · NeurIPS 2020]

This paper has received 3 very favorable reviews, describing a rigorous and well written paper proposing a new method for data-driven modeling of dynamical systems, an important field of research. The reviewers laud novelty, applicability to a broader class of problems, empirical results on a variety of problems, and excellent positionning wrt the state of the art. The AC concurs.